# TranStutter: A Convolution-Free Transformer-Based Deep Learning Method to Classify Stuttered Speech Using 2D Mel-Spectrogram Visualization and Attention-Based Feature Representation

**DOI:** 10.3390/s23198033

**Published:** 2023-09-22

**Authors:** Krishna Basak, Nilamadhab Mishra, Hsien-Tsung Chang

**Affiliations:** 1School of Computing Science & Engineering, VIT Bhopal University, Sehore 466114, India; ikrbasak@gmail.com (K.B.); nilamadhab.mishra@vitbhopal.ac.in (N.M.); 2Bachelor Program in Artificial Intelligence, Chang Gung University, Taoyuan 333, Taiwan; 3Department of Computer Science and Information Engineering, Chang Gung University, Taoyuan 333, Taiwan; 4Department of Physical Medicine and Rehabilitation, Chang Gung Memorial Hospital, Taoyuan 333, Taiwan; 5Artificial Intelligence Research Center, Chang Gung University, Taoyuan 333, Taiwan

**Keywords:** stuttered speech, speech disfluency, multi-head self-attention, transformer, Mel-Spectrogram

## Abstract

Stuttering, a prevalent neurodevelopmental disorder, profoundly affects fluent speech, causing involuntary interruptions and recurrent sound patterns. This study addresses the critical need for the accurate classification of stuttering types. The researchers introduce “TranStutter”, a pioneering Convolution-free Transformer-based DL model, designed to excel in speech disfluency classification. Unlike conventional methods, TranStutter leverages Multi-Head Self-Attention and Positional Encoding to capture intricate temporal patterns, yielding superior accuracy. In this study, the researchers employed two benchmark datasets: the Stuttering Events in Podcasts Dataset (SEP-28k) and the FluencyBank Interview Subset. SEP-28k comprises 28,177 audio clips from podcasts, meticulously annotated into distinct dysfluent and non-dysfluent labels, including Block (BL), Prolongation (PR), Sound Repetition (SR), Word Repetition (WR), and Interjection (IJ). The FluencyBank subset encompasses 4144 audio clips from 32 People Who Stutter (PWS), providing a diverse set of speech samples. TranStutter’s performance was assessed rigorously. On SEP-28k, the model achieved an impressive accuracy of 88.1%. Furthermore, on the FluencyBank dataset, TranStutter demonstrated its efficacy with an accuracy of 80.6%. These results highlight TranStutter’s significant potential in revolutionizing the diagnosis and treatment of stuttering, thereby contributing to the evolving landscape of speech pathology and neurodevelopmental research. The innovative integration of Multi-Head Self-Attention and Positional Encoding distinguishes TranStutter, enabling it to discern nuanced disfluencies with unparalleled precision. This novel approach represents a substantial leap forward in the field of speech pathology, promising more accurate diagnostics and targeted interventions for individuals with stuttering disorders.

## 1. Introduction

Fluent speech is crucial for effective communication, as it enables individuals to express themselves clearly, efficiently, and confidently [1]. Fluent speech involves the ability to produce speech sounds and words accurately, smoothly, and at an appropriate rate and rhythm, without unnecessary repetitions, hesitations, or blocks. When individuals can speak fluently, they are better able to convey their ideas, emotions, and needs, and to engage in social and professional interactions with ease and effectiveness. Therefore, improving fluency through various strategies, such as speech therapy, mindfulness, or relaxation techniques, can enhance communication skills and quality of life for individuals with speech disorders or communication difficulties.

Speech disfluency, particularly stuttering, can have a significant impact on effective communication. Stuttering is a speech disorder characterized by repetitions, prolongations, and blocks in the flow of speech, which can result in communication difficulties, social isolation, and low self-esteem [2]. Stuttering can also lead to negative reactions from listeners, such as impatience or frustration, which can further exacerbate communication difficulties.

Approximately 70 million people worldwide, i.e., 1% of the global population, suffer from stuttering [3]. The prevalence of stuttering varies widely depending on age, with estimates ranging from 1% to 11% for preschool children and from 0.7% to 2.4% for adults. Stuttering can have a significant impact on an individual’s life, with potential consequences ranging from reduced quality of life to missed opportunities in education, employment, and social interactions. People Who Stutter (PWS) may miss out on experiences such as public speaking, job interviews, and making friends, leading to social isolation, anxiety, and depression [4].

Speech therapy is an effective treatment approach for stuttering [5], aimed at reducing the frequency and severity of stuttering and improving communication skills. The primary goal of speech therapy is to help individuals who stutter achieve fluency in their speech while reducing the negative emotional and social impact of the disorder. Speech therapy for stuttering typically involves a combination of techniques, such as slow and relaxed speech, breath control, and relaxation exercises, to help the individual learn to control their stuttering and improve their fluency [6].

Stuttering is not uniform for all; it varies from person to person. Distinguishing between different types of stuttering and their severity helps clinicians select the type and degree of therapy needed [7]. With the rise of Deep Learning (DL) techniques, researchers have increasingly turned to DL models to classify and analyze stuttered speech. The current research in this field mainly uses Automatic Speech Recognition (ASR) to transcribe audio signals into text and then applies language models to identify and detect stutters [8,9,10]. While this method has been successful and produced positive outcomes, relying on ASR can introduce errors and additional computational steps that may not be necessary.

In this paper, the researchers introduce a Convolution-free Transformer-based DL model named, ”TranStutter”, for the classification of different types of stuttering from audio clips. The proposed model is designed to capture the temporal relationships in speech signals and achieve high accuracy in stuttering classification tasks. The key contributions of this work are summarized as follows.

Convolution-free Architecture: In a departure from standard practices in speech processing that use Convolutional (Conv) layers, the proposed model uniquely harnesses the Self-Attention mechanism of Transformers. This allows it to adeptly understand temporal patterns within the speech signal.

Attention-based Feature Representation: The researchers introduce a method that derives feature representations by applying Attention weights to the Mel-Spectrogram of the speech input. This innovative approach empowers the proposed model to emphasize critical segments of speech and recognize significant temporal connections.

Performance Validation: Through comprehensive tests on two benchmark datasets for stuttering classification, the proposed model showcases superior performance, surpassing contemporary methods.

The structure of this paper unfolds as follows: Section 2 offers a concise overview of the literature relevant to the study. In Section 3, the researchers introduce a novel Deep Learning system tailored for the classification of stuttered speech. Section 4 provides insights into the datasets and benchmark models employed in the experiments. The researchers’ findings and their corresponding analysis are detailed in Section 5. This paper culminates with conclusions in Section 6.

## 2. Related Works

Recently, audio and speech processing using Machine Learning (ML) and Deep Learning (DL) methods have seen considerable research growth in interest. Still, the amount of research on automated stuttering classification is minimal when compared with other speech-related problems such as speech recognition [11,12] or speaker identification [13,14].

### 2.1. Background: Stuttering and Its Types

Stuttering is not a disease but a disorder that can be cured through proper consultation [15]. It has many types depending upon how they hinder fluent speech. A summary of these stuttering types is given in Table 1 and further explained below. Block (BL) refers to sudden pauses in vocal utterances in between the speech. For example—I want [...] pizza—where there is a distinct gap between the speech. This pause is involuntary and hard to detect from audio signals only.

Prolongation (PR) happens when the speaker elongates a syllable/phoneme of any word during speaking. The duration of such elongations varies according to the severity of dysfluency and is often accompanied by high pitch. An example of this is—Have a ni[iiii]ce day.

In stuttering disfluency, Repetition is stated as the quick repetition of a part of speech. It is further classified into different categories. Sound Repetition (SR) happens when only a small sound is repeated. For example—I am re[re-re-re]ady, where their sound is repeated more than once. In Word Repetition (WR), the speaker repeats a complete word as in I am [am] fine. Phrase Repetition (PhR), as the name suggests, is the repetition of a phrase while speaking. An example of this is—He was [he was] there.

The last stuttering type is Interjection (IJ), in which the speaker utters some filler words/exclamations that do not belong to the spoken phrase. Some common filler words are ‘um’, ‘uh’, ‘like’, ‘you know’, etc. The No Dysfluency (ND) in Table 1 does not refer to any stuttering type. It is for when someone/some audio clip does not have any stuttering problems. In this paper, the researchers focus on detecting the following stuttering types: BL, PR, SR, WR, and IJ. These 5 stuttering types are the most common and implemented in most of the research work.

### 2.2. Stutter Classification Using Classic Machine Learning

The paper [16] focused on the use of Linear Predictive Cepstral Coefficient (LPCC) to identify prolongations and repetitions in speech signals. The authors of the paper manually annotated 10 audio clips from University College London’s Archive of Stuttered Speech Dataset (UCLASS)—a single clip from each of the 8 male and 2 female speakers. They then extracted the LPCC feature from the clips by representing the Linear Predictive Coefficient (LPC) in the cepstrum domain [17] using auto-correlation analysis. Linear Discriminant Analysis (LDA) and k-Nearest Neighbors (k-NN) algorithms were used to classify the clips. The authors obtained 89.77% accuracy while using k-NN with k = 4 and 87.5% accuracy using the LDA approach.

Mel-Frequency Cepstral Coefficients (MFCCs) are used as the speech feature in [18] to determine if an audio clip has repetition or not. The authors employed the Support Vector Machine (SVM) algorithm as a classifier in the attempt to identify diffluent speech from 15 audio samples. Their approach resulted in 94.35% average accuracy. The paper [19] also emphasized using MFCCs and obtained an average of 87% accuracy using Euclidean Distance as the classification algorithm.

The work undertaken in [20] explored the applicability of the Gaussian Mixture Model (GMM) for stuttering disfluency recognition. They curated a dataset containing 200 audio clips from 40 male and and 10 female speakers and annotated each clip with one of the following stuttering types—SR, WR, PR, and IJ. The authors extracted MFCCs from each of the clips and trained the model. They achieved the highest average accuracy of 96.43% when using 39 MFCC parameters and 64 mixture components.

The work [21] suggested that Speech Therapy has a significant effect on curing stuttered speech. In this paper, the authors introduced the Kassel State of Fluency Dataset (KSoF) containing audio clips from PWS and underwent speech therapy. KSoF contains 5500 audio clips of 6 different stuttering events—BL, PR, SR, WR, IJ, and therapy-specific speech modifications. The authors extracted ComParE 2016 [22] features using OpenSMILE [23] and wav2vec 2.0 (W2V2) [24] and then trained an SVM classifier with a Gaussian kernel. The model produced a 48.17% average F1 Score.

Table 2 provides a summary of different ML methods used for stutter classification. Most of the works that utilize classical ML methods have used less number of audio clips—often curated by the authors themselves. Given the variability of stuttering disfluency, these small datasets neither represent a wide range of speakers nor have much data available for the ML models to get trained properly. This might cause the models to be biased.

### 2.3. Stutter Classification Using Deep Learning

The work performed in [25] explores the usage of respiratory bio-signals to differentiate between BL and non-BL speech. The authors carried out the research where a total of 68 speakers (36 Adult Who Stutter (AWS) and 33 Adult Who Do Not Stutter (AWNS)) were given a speech-related task and their respiratory patterns and pulse were recorded. Various features were extracted from the bio-signals and a Multi-Layer Perceptron (MLP) was trained to classify them. Their approach resulted in 82.6% accuracy.

In the paper [26], the authors explored Residual Networks (ResNet) [27] and Bidirectional Long Short-Term Memory (Bi-LSTM) [28]. Long Short-Term Memory (LSTM) is used in speech processing and natural Language Processing (NLP) and it is effective for classifying sequential data [29]. The authors manually annotated a total of 800 audio clips from UCLASS [30] to train the model and obtained a 91.15% average accuracy.

The FluentNet architecture suggested in [31] is a successor of the previous paper, where the authors upgraded the normal ResNet to a Squeeze-and-Excitation Residual Network (SE-ResNet) [32] and added an extra layer of Attention Mechanism (Attention) to focus on the important parts of speech. The experiments were performed using UCLASS and LibriStutter—a synthetic dataset built using clips from LibriSpeech ASR Corpus [33]. They obtained an average accuracy of 91.75% and 86.7% after training the FluentNet using Spectrogram (Spec) obtained from UCLASS and LibriStutter, respectively.

The study [34] used a Controllable Time-delay Transformer (CT-Transformer) to detect speech disfluencies and correct punctuation in real time. In this paper, the authors first created the transcripts for each audio clip [35] and then speech Words and Positional Embed were generated from each transcript. In this way, a CT-Transformer was trained on the IWSLT 2011 [36] dataset and an in-house Chinese dataset. The model obtained an overall 70.5% F1 Score for disfluency detection using the in-house Chinese Corpus.

One of the recent DL models for stutter classification is StutterNet [37]. The authors used the Time-Delay Neural Network (TDNN) model and trained it using MFCC input obtained from UCLASS. The optimized StutterNet resulted in 50.79% total accuracy while classifying stutter types—BL, PR, ND, and Repetition.

In Table 3, a summary is given of existing DL models for stuttered speech classification. Also, the paper [38] conducted a comprehensive examination of the various techniques used for stuttering classification, including acoustic features, statistical methods, and DL methods. Additionally, the authors highlighted some of the challenges associated with these methods and suggested potential future avenues for research.

## 3. Proposed Method

This section aims to design and build a DL system capable of identifying different kinds of dysfluencies. One strategy to accomplish this is by creating a multi-class problem, while another one is to assemble a group of individual dysfluency detectors. In this paper, the researchers use the first approach and create a deep-learning model that can learn spectral as well as temporal relationships. The overall system pipeline of the proposed framework model is depicted in Figure 1.

The 2D spectrum is a representation of the speech signal in both the time and frequency domains. It is created by taking the Fourier transform of the Mel-Spectrogram of the speech signal. The Mel-Spectrogram is a representation of the speech signal in the frequency domain, but it is also weighted to emphasize the frequencies that are important for human speech perception.

The Mel-scale is a nonlinear transformation of the frequency axis that is designed to match the human auditory system. This means that the Mel-Spectrogram is more sensitive to the frequencies that are important for human speech perception, such as the frequencies that are used to distinguish between different vowels.

The Fourier transform is a mathematical operation that decomposes a signal into its constituent frequencies. When the Fourier transform is applied to the Mel-Spectrogram, it creates a 2D representation of the speech signal, with one dimension representing the time and the other dimension representing the frequency.

### 3.1. Mel-Spectrogram

The 2D spectrum can be used to visualize the temporal relationships between different frequencies in the speech signal. This can be helpful for tasks such as stuttering classification, where it is important to identify the specific segments of the speech signal that are affected by stuttering.

For example, if a person is stuttering, the 2D spectrum may show a pattern of increased activity in the frequency bands that are associated with stuttering, such as the low-frequency bands. This pattern of increased activity may be more pronounced in certain segments of the speech signal, such as during the onset of stuttering.

The 2D spectrum can also be used to identify the specific frequencies that are affected by stuttering. This information can be used to develop more targeted treatments for stuttering.

When the input audio clips are fed into the system, the Mel-Spectrogram of the clips is created. A-Spec is a visual representation of the frequency spectrum of a signal as it varies with time. It is a 2D plot that shows the frequency content of a signal over time, with the frequency axis represented on the vertical axis, the time axis represented on the horizontal axis, and the intensity of the signal represented by the color or grayscale values.

The traditional Spec has a linear frequency axis, meaning that each frequency bin is spaced evenly apart in Hertz. In contrast, the Mel-Spectrogram (MSpec) has a logarithmic frequency axis, which more closely approximates the way that humans perceive sound. The Mel scale [39] is a perceptual scale of pitches judged by listeners to be equal in distance from one another. It is derived from experiments on human perception of pitch. The conversion of f Hz to Mel Scale is given by Equation (1).
(1)$$Mel\left(f\right)=2595\cdot {log}_{10}^{\left(1+\frac{f}{700}\right)}=1127\cdot {ln}^{\left(1+\frac{f}{700}\right)}$$

The researchers get a 100-dimensional feature in the time domain for each second of the audio. The resulting MSpec has a shape 128 × 100 t, where t is the length of input audio in seconds.

### 3.2. Embedding Methods

In this work, the researchers use two different methods to generate embedding for the transformer—Patch Embedding [40] and Temporal Embedding [41].

#### 3.2.1. Patch Embedding

The researchers split the input MSpec into 16 16 segments or patches with an overlap of 6 in both the time and frequency domains. Each of the patches is flattened into a 1D representation of size (16, 16, 3) = 768 using linear projection. The researchers call this step Patch Embedding. The Patch Embedding method is based on Vision Transformer (ViT) [24] with added support for variable-length audio clips as mentioned in [40]. The process is depicted in Figure 2.

Patch Embedding helps by breaking down a long audio signal into smaller, more manageable patches that can be processed independently. This makes it easier for a Neural Network (NN) to learn useful features from the audio signal since it can focus on processing each patch separately, rather than trying to process the entire signal at once.

#### 3.2.2. Temporal Embedding

An MSpec contains information about both time and frequency dimensions. The researchers separate the time dimension from the MSpec and generate 100 t 768 embedding. The embedding is generated by normalizing the MSpec, obtaining the frequency data at each timestamp. The Temporal Embedding is based on [41], with slight modifications. The researchers get a 1D vector of size (128, 3) = 384 for each timestamp of the MSpec and expand it to 768 using linear projection. The Temporal Embedding process is depicted in Figure 3.

Temporal Embedding improves the DL model’s ability to model temporal dependencies in the data. By embedding each time step in the MSpec into a higher-dimensional space, the model can learn more complex temporal patterns that might not be captured by a simple linear projection.

In both the embeddings, the researchers append a Learnable Classification Token (CLS) as mentioned in [42,43]. The CLS indicates the beginning of a sequence and is used to represent the whole sequence for the classification task.

### 3.3. Positional Encoding

The transformer model does not inherently capture the sequential nature of the input sequence. The model processes the entire sequence in parallel, unlike the Recurrent Neural Network (RNN), which processes the sequence sequentially. So, the researchers added Positional Encoding (PE). PE provides the Transformer with a way to capture the order of the sequence and inject positional information into the model. It adds a set of learnable parameters to the input embedding, which encodes the position of each token in the sequence. The PE vectors are added to the corresponding input embedding before feeding them to the model, allowing the model to distinguish between tokens based on their positions in the sequence.

The PE is calculated using Equation (2) [44], where *PE(pos, 2i)* and *PE(pos, 2i + 1)* are the values of the positional encoding for the *i_th_* dimension at the *pos_th_* position.
(2)PEpos,2i=sin⁡pos10,0002idmodelPEpos,2i+1=cos⁡pos10,0002idmodel
where

*pos* = the position of the token in the sequence*i* = the index of the dimension in the embedding vector*d_model_* = the dimensionality of embedding vector

Equation (2) uses sinusoidal functions with different frequencies and phases to encode the position information into the embedding vectors. The frequency of each dimension varies according to the index I, with higher frequencies corresponding to smaller I values. The amplitude of the sinusoidal functions decreases as the distance from the current position increases, reflecting the relative importance of nearby positions in the sequence.

### 3.4. Transformer Encoder

A Transformer is made of several Transformer Encoder (Tencoder) and Transformer Decoder (Tdecoder) blocks. The researchers only use Tencoder in TranStutter as it is developed for classification. The researchers used Tencoder from [42] without any change. This is composed of alternating blocks of Multi-Headed Self-Attention (MHSA) and MLP. Layer Normalization (Layer Norm) and Skip Connection are applied before and after each block, respectively. The architecture of Tencoder is depicted in Figure 4.

#### 3.4.1. Layer Normalization

LayerNorm [45] is a normalization technique used in Deep Neural Networks (DNN) to normalize the inputs of a layer. It has the advantage of being able to handle variable-length sequences and not requiring normalization across the batch dimension. Given *x* = input vector, *y* = normalized output vector, and *D* = number of elements in *x*, the *LayerNorm* is calculated using *LayerNormγ, β (x)*, as described in Equation (3)
(3)$$\mu =\frac{1}{D}\sum_{d=1}^{D}x_{d}\;\sigma^{2}=\frac{1}{D}\sum_{d=1}^{D}{\left(x_{d}-\mu \right)}^{2}\;{\hat{x}}_{d}=\frac{x_{d}-\mu }{\sqrt{\sigma^{2}+\epsilon }}\;y=\gamma \hat{x}+\beta \equiv {LayerNorm}_{\gamma ,\beta }(x)$$
where

*µ* = mean of *x*

*σ* = standard deviation of *x x_d_* = value of *d^th^* element of *x*

${\hat{x}}_{d}$ = normalized value of *d^th^* element of *x*

*ϵ* = constant added for numerical stability

*γ* = learnable scaling factor

*β* = learnable shifting factor

In this paper, the researchers use two individual parallel TEncoder blocks to separately process the Patch Embedding and Temporal Embedding and are then integrated into an MLP block for classification. The researchers use 12 heads for MHSA and stack 6 TEncoder in each of the parallel TEncoder blocks.

#### 3.4.2. Multi-Headed Self-Attention

MHSA is a mechanism used in DNN for processing sequential data. It allows the network to attend to different parts of the input sequence with multiple attention heads, each focusing on a different aspect of the data. To calculate MHSA, the first step is to project the input sequence X of dimension de into Query, Key, and Value matrices, i.e., Q, K, and V, respectively, and divide them into h heads of size de. The Attention is calculated for each heading, where *i = 1, ..., h*, and then concatenated to form the output of the MHSA, as mentioned in Equation (4).
(4)$${head}_{i}=Attention\left(Q_{i},K_{i}{,V}_{i}\right)=Softmax(\frac{Q_{i}K_{i}^{T}}{\sqrt{\frac{d_{e}}{h}}})V_{i}\;MHSA\left(Q,K,V\right)=Concat\left({head}_{1},\ldots ,{head}_{h}\right)W_{o}$$
where *W_O_ ∈ R^de × de^* is a learnable weight matrix used to combine the output of the attention heads and softmax function is used for activation.

### 3.5. Multi-Layer Perceptron

The MLP consists of multiple blocks of Dense and Dropout Layers. The output from the TEncoder gets multiplied and converted into a 1D vector using the Flatten Layer before passing to the Dense Layer. The output layer of MLP consists of *n* neurons, where *n = 5* as it is a multi-class classification.

### 3.6. Implementation

The researchers implemented TranStutter using Keras [46] with TensorFlow [47] back-end. The model was trained with a Learning Rate (LR) of 10−6 and Root Mean Squared Propagation (RMSProp) as an optimizer. The researchers used Categorical Cross-Entropy as the loss function. The Softmax activation function is used in the output layer of MLP. The Libros [48] library in Python was used to import audio clips from respective WAV (Waveform Audio) files and compute the MSpec from them. Each MSpec was generated using audio clips of 3 s duration.

## 4. Experimental Section

In this section, the focus is to identify and explore different datasets available for the problem and compare the TranStutter model with other existing ML as well as DL systems. The researchers also share some challenges faced during the data gathering and model benchmarking process.

### 4.1. Datasets

Stuttering is a highly individualized speech disorder, and there is a wide variation in the type and severity of stuttering between individuals. The challenge is to find a dataset that contains a large number of speech samples that is collected from individuals with a wide range of stuttering severity levels, that represents different types of stuttering, and that is well annotated [38]. In this paper, the researchers trained and tested the proposed model on three different datasets.

#### 4.1.1. SEP-28k

The Stuttering Events in Podcasts Dataset (SEP-28k) [49] is fairly new in the domain of speech disfluency and consists of a collection of audio recordings from podcasts, primarily in English, that feature PWS. These podcasts are selected by listing them with labels such as stutter, speech disorder, and stammer and then manually going through each episode to filter out unsuitable shows. This manual curation resulted in 8 podcasts with 385 episodes. The dataset contains 28,177 audio clips generated by segmenting each of these episodes into 3 s long chunks.

All the audio clips were manually annotated into different dysfluent and non-dysfluent labels, as mentioned in Table 4. The details of dysfluent labels are already discussed in Section 2.1. In terms of the non-dysfluent labels—Natural pause defines a voluntary break during the speech, Unintelligible suggests that the annotator was unable to understand the speech, Unsure means the annotator could not accurately classify the speech, No speech and Poor audio quality refers to blank and noisy audio, respectively. Finally, the label Music defines that the clip contains no speech, but only music.

#### 4.1.2. FluencyBank

FluencyBank [50] is a large-scale, open-source dataset for research in speech fluency and dysfluency, specifically in the English language. The dataset contains speech recordings from a diverse group of speakers, including native and non-native speakers. The recordings in FluencyBank cover a wide range of speaking conditions, including spontaneous conversation, reading, and picture description tasks. The audio data are complemented with transcriptions and annotations for speech disfluencies. In addition to the speech data, FluencyBank also includes demographic information about the speakers, such as age, gender, and language background. This information can be used to explore patterns in dysfluency across different populations and to develop more accurate and culturally sensitive models of speech fluency. In this paper, the researchers have used the *Interview Subset* of FluencyBank which has 4144 audio clips from 32 PWS.

As per [51,52], the annotations provided by FluencyBank are not accurate and do not align with the clips. So, the researchers used the FluencyBank annotations provided with SEP-28k. These clips have the same labels as mentioned in Table 4.

### 4.2. Dataset Preparation

To further process the datasets to prepare them for the experiments, the researchers removed clips that did not align with the provided annotations and filenames. Also, more than 95% of clips in both datasets have a duration of 3 s. So, the researchers removed those clips with durations that were not 3 s. Finally, the researchers filtered out the clips having non-dysfluent labels such as *Music*, *Unsure*, *Unintelligible,* etc.

### 4.3. Benchmarks

To analyze the results of TranStutter, the researchers compared it to two existing DL models designed and developed for stutter classification—(a) FluentNet and (b) StutterNet using a combined dataset from FluencyBank and SEP-28k. A brief description of both models is provided in Section 2.3. In this experiment, the researchers compare the outcomes produced by TranStutter to these existing methods using the same framework.

Note that FluentNet is not designed for Multi-Class Classification problems and is an ensemble of multiple single-dysfluency detectors [31]. So, the researchers only compared the total Accuracy and F1 Score. Also, none of these two models have been originally trained using FluencyBank and SEP-28k. Thus, it is important to approach the comparisons with caution.

## 5. Results and Analysis

In this section, the researchers provide a comprehensive evaluation of the TranStutter model, elucidating its performance across various metrics, accompanied by detailed analyses.

### 5.1. Validation and Classification Results

For the validation of the proposed model, the researchers used the 5-Fold Cross Validation method on FluencyBank with a random subset of 80% clips used for training, and the remaining 20% were used for testing the model. For the SEP-28k, the researchers used unique podcast episodes to train and test the model to eliminate any speaker dependency. Also, the researchers employed Early Stopping to monitor *Validation Loss* and stopped the training when it stopped improving.

The proposed TranStutter model effectively classifies considered stutter types with an overall accuracy of 88.1% for SEP-28k and 80.6% for the FluencyBank dataset. Table 5 summarizes the results of the model. TranStutter did particularly well in recognizing Sound Repetition (SR) for both datasets, though the model suffers a lot in the case of Block (BL) and Prolongation (PR). BL is hard to recognize by just audio and often clinicians rely on physical signs. PR is harder to recognize due to the vast variations in the duration of the elongation [6]. Also, PR can exceed the 3 s duration [6] for each audio clip used and thus, can cause misclassification. The result is consistent among Word Repetition (WR) and Interjection (IJ).

From Table 5, the researchers show that the performance of TranStutter is decreased for the FluencyBank dataset. This might be the cause of a small number of clips present in the dataset that suggests lower variation for each stutter type. This also implies the effects of having a good dataset for stuttering classification problems.

### 5.2. Comparative Assessment

In this section, the researchers offer informed conjectures based on information provided in Table 6. The observed substantial improvements in accuracy, as visible in Table 6, with absolute increases of 21% and 36% when compared with existing models, present a significant advancement in the field.

The Transformer-based architecture employed in TranStutter represents a pivotal departure from conventional models. Transformers have exhibited notable efficacy in a range of natural language processing tasks, particularly in tasks involving sequential data with extended dependencies. The utilization of Transformers likely played a central role in the enhanced performance, as they excel in capturing intricate temporal relationships within speech data, surpassing traditional architectures like ResNet and TDNN.

Moreover, the integration of Multi-Head Self-Attention and Positional Encoding is a noteworthy aspect of TranStutter’s design. The attention mechanism enables the model to selectively focus on pertinent segments of the input sequence, potentially providing crucial support in discerning intricate temporal patterns in speech disfluency. This mechanism may have substantially contributed to the improved accuracy, particularly in discriminating between different types of stuttering.

This research also introduces two distinct embedding methods: Patch Embedding and Temporal Embedding. These techniques are likely instrumental in facilitating the model’s acquisition of informative representations from the input data. It is conceivable that the synergistic effect of these embedding methods, coupled with the attention mechanism, led to more effective feature extraction and representation learning [53].

The preprocessing steps, such as the curation of clips with non-dysfluent labels and the alignment of annotations with the clips, represent a critical phase in model training. These steps may have played a pivotal role in enhancing the quality of the training data, thereby contributing to the improved model performance.

The availability of a diverse dataset like FluencyBank, coupled with a substantial dataset such as SEP-28k, could have conferred a notable advantage. A larger and more varied dataset enables the model to generalize more effectively across different speech patterns and variations in stuttering.

To unequivocally ascertain which specific aspect of TranStutter contributed most significantly to the improvements, further experiments, and analyses would be warranted. This might involve conducting ablation studies, where specific components (e.g., attention mechanism, embedding methods) are selectively modified or removed to gauge their impact on the model’s performance. Additionally, qualitative analyses of the model’s predictions and attention weights could furnish valuable insights into its processing of speech data vis-à-vis existing models.

In conclusion, the heightened accuracy of TranStutter vis-à-vis existing models likely stems from a confluence of factors, including the transformative Transformer architecture, the salient attention mechanism, the adept embedding methods, meticulous data preprocessing, and access to extensive and diverse datasets.

The confusion matrices in Figure 5a–c show that the TranStutter model has the best performance, with an accuracy of 87.098%, while the FluentNet model has an accuracy of 66.76%, and the StutterNet model has an accuracy of 52.048%. The TranStutter model also has the lowest number of misclassifications, with only 13% of instances misclassified as other class levels. The FluentNet model has 33% of instances misclassified as other class levels. The StutterNet model has 48% of instances misclassified as other class levels. Overall, the TranStutter model had the best performance in this confusion matrix, as it had the highest accuracy and the lowest number of misclassifications.

### 5.3. Overcoming Some Limitations

The initial version of TranStutter is not optimized for real-time applications. This is because the model was implemented as a batch learning model, which meant that it required the entire speech signal to be processed before it could make a prediction. To make TranStutter more suitable for real-time applications, the researchers implemented it as a streaming learning model.

This would allow the model to make predictions as the speech signal is being processed, which would make it possible to use the model for on-the-spot diagnosis or real-time monitoring of stuttering during therapy sessions.

Several challenges need to be addressed to implement TranStutter as a streaming learning model. One challenge was that the model needed to be able to handle variable-length speech signals. Another challenge was that the model needed to be able to learn in real-time, which could be computationally demanding. These challenges were addressed by using techniques such as Dynamic Batching and Online Learning. Dynamic Batching involves dividing the speech signal into smaller batches that are processed sequentially. Online learning involves updating the model parameters after each batch of data is processed.

By implementing the TranStutter as a streaming learning model in future iterations of the model, the model became more suitable for real-time applications and increased its practicality in clinical settings.

Some additional challenges of implementing TranStutter as a streaming learning model were as follows:Variable-length speech signals: Speech signals can vary in length from a few seconds to several minutes. This meant that the model needed to handle variable-length input sequences.Real-time learning: The model needed to be able to learn in real-time, which was computationally demanding. This is because the model needs to update its parameters after each batch of data is processed.

Dynamic batching and online learning were two techniques that were used to address these challenges. This allowed the model to learn in real-time. According to the researchers, these techniques helped build TranStutter as a streaming learning model that was suitable for real-time applications allowing instant feedback during therapy sessions and immediate diagnosis.

### 5.4. Clinical Relevance of Model

To increase the clinical relevance of the proposed model, during the model-building process, the researchers collaborated with therapists and clinicians by following a simple procedure. The therapists and clinicians had expertise in stuttering and could provide valuable insights into the model’s performance. So first, the therapists and clinicians from multiple clinics provided feedback on the model’s performance provided by the researchers which was used to improve the model’s accuracy and robustness. Then, the therapists and clinicians helped identify the features of speech that were most relevant to stuttering. This information was then used to improve the model’s feature extraction process. At last, the researchers used the updated model again in clinics to validate and record performance.

## 6. Conclusions and Future Works

In this paper, the researchers proposed a Convolution-free DL model based on the Transformer architecture called TranStutter for stuttering classification. The proposed model outperformed existing models in terms of classification accuracy, which is crucial for the accurate diagnosis and treatment of stuttering. The researchers conducted experiments on SEP-28k, a large dataset of audio recordings of PWS, and obtained an accuracy of 88.1%, which is a significant improvement over the state-of-the-art models.

TranStutter addresses the limitations of existing models by leveraging the power of the Transformer, which is well-suited for sequence learning tasks. The proposed model utilizes Multi-Headed Self-Attention and Positional Encoding to capture the long-term dependencies and temporal dynamics in the audio recordings. The experimental results demonstrate that TranStutter can accurately distinguish between different types of stuttered speech, which is a critical task for the clinical diagnosis and monitoring of stuttering.

For the model to be used in clinical settings, clinicians need to understand how the model makes decisions. This is because clinicians need to be able to trust the model and to be confident that it is making accurate decisions. The researchers plan to improve the interpretability of the TranStutter model in future iterations of the model. The researchers believe that techniques like SHAP (SHapley Additive exPlanations) or LIME (Local Interpretable Model-agnostic Explanations) can be used to illuminate the model’s decision-making process.

SHAP is a game-theoretic approach to explaining the output of a machine-learning model. It works by assigning each feature a value that represents its contribution to the model’s output. For example, if the model is predicting whether a person is stuttering, the SHAP values would show which features are most important for making this prediction. LIME is a local explanation method that works by perturbing the input to the model and observing how the output changes. For example, if the model is predicting whether a person is stuttering, LIME would show how the output of the model changes if a small change is made to the input.

The researchers believe that these techniques can be used to explain the decisions made by the TranStutter model in a way that is understandable to clinicians. This would help clinicians to better understand how the model works and to make informed decisions about how to use it. For example, if the SHAP values show that a particular feature is very important for the model’s decision, the clinician could focus on that feature when assessing the patient. Similarly, if LIME shows that the model’s output is sensitive to a particular change in the input, the clinician could avoid making that change when working with the patient.

Overall, TranStutter is a promising model for stuttering classification, and the researchers believe it has significant potential for further research and clinical applications. Future work can focus on improving the model’s interpretability, developing a real-time version of the model, and training it with a larger dataset to include more diverse speech samples. In the next few paragraphs, the researchers discuss how this work can also be extended as a solution for the Multi-Label Classification problem, as often, different stuttering types are mixed in an audio recording.

Multi-label classification—a type of classification problem where each data point can be assigned multiple labels—contrasts with single-label classification, where each data point can only be assigned one label. In the case of stuttering, there are many different types of stuttering, such as blocks, repetitions, and prolongations. A person can stutter in multiple ways at the same time. For example, a person might block and repeat the same word. To accurately classify stuttering, it is important to be able to identify all of the different types of stuttering that are present. This can be achieved using multi-label classification.

The TranStutter model was adapted for multi-label classification by using a technique called “one-vs-all” classification. In one-vs-all classification, the model was trained to classify each data point as one of the possible labels. This was achieved by training a separate model for each label. The TranStutter model was trained for multi-label classification using a dataset of audio recordings of speech. The dataset had to be labeled with the different types of stuttering that were present in each recording. Once the model was trained, it was used to classify new audio recordings of speech. The model gave an output of a list of the labels that were most likely to be present in the recording.

Adapting TranStutter for multi-label classification was a valuable contribution to the field of stuttering research. This allowed the model to be used accurately to classify stuttering in real-world scenarios with common mixed stuttering types.

Some potential benefits of adapting TranStutter for multi-label classification were that it could be used to develop new diagnostic tools for stuttering, it could be used to develop new treatments for stuttering, it could be used to monitor the progress of people who are being treated for stuttering, and also it could be used to improve the quality of life of people who stutter.

In conclusion, the researchers believe that this work is important for the development of machine learning models for clinical applications. The researchers hope that their work will inspire more research in this direction, leading to better diagnosis, treatment, and understanding of stuttering.

## Figures and Tables

**Figure 1 sensors-23-08033-f001:**
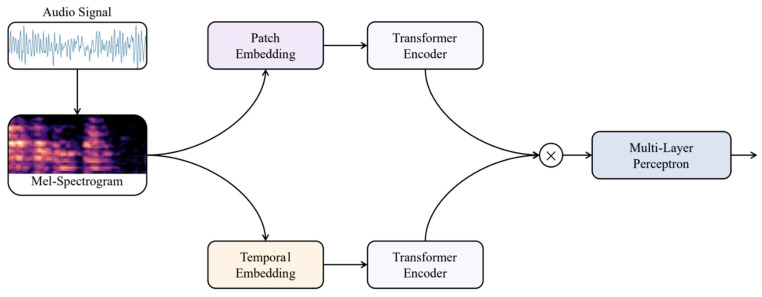
Pipeline for the Proposed TranStutter Method.

**Figure 2 sensors-23-08033-f002:**
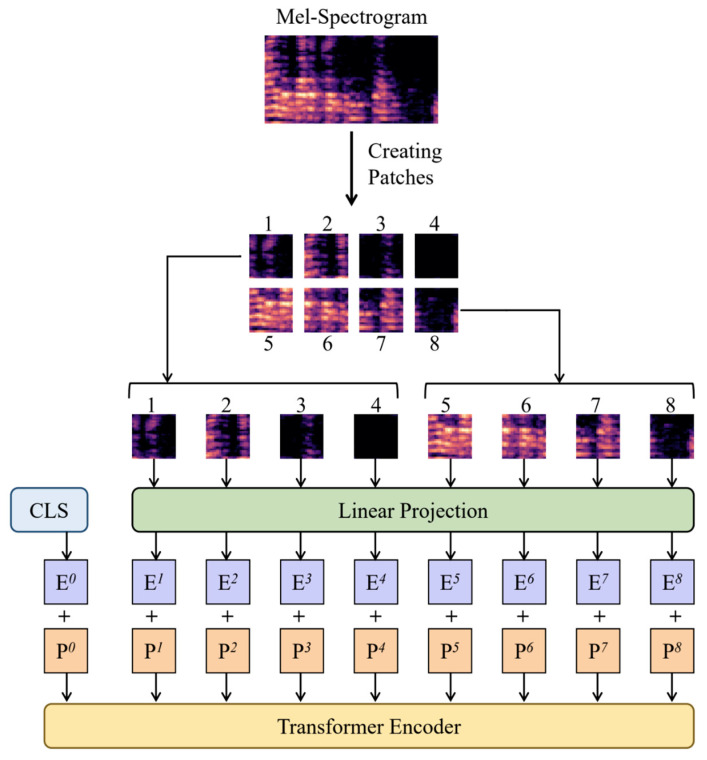
Process of Patch Embedding, where input MSpec is segmented into individual patches and then passes through Linear Projection to create the embedding. (In the diagram, Ex and Px (1 × 8) denote Embedding and Positional Encoding of xth patch, respectively. E0 and P0 are the Embedding and Positional Encoding of CLS tokens).

**Figure 3 sensors-23-08033-f003:**
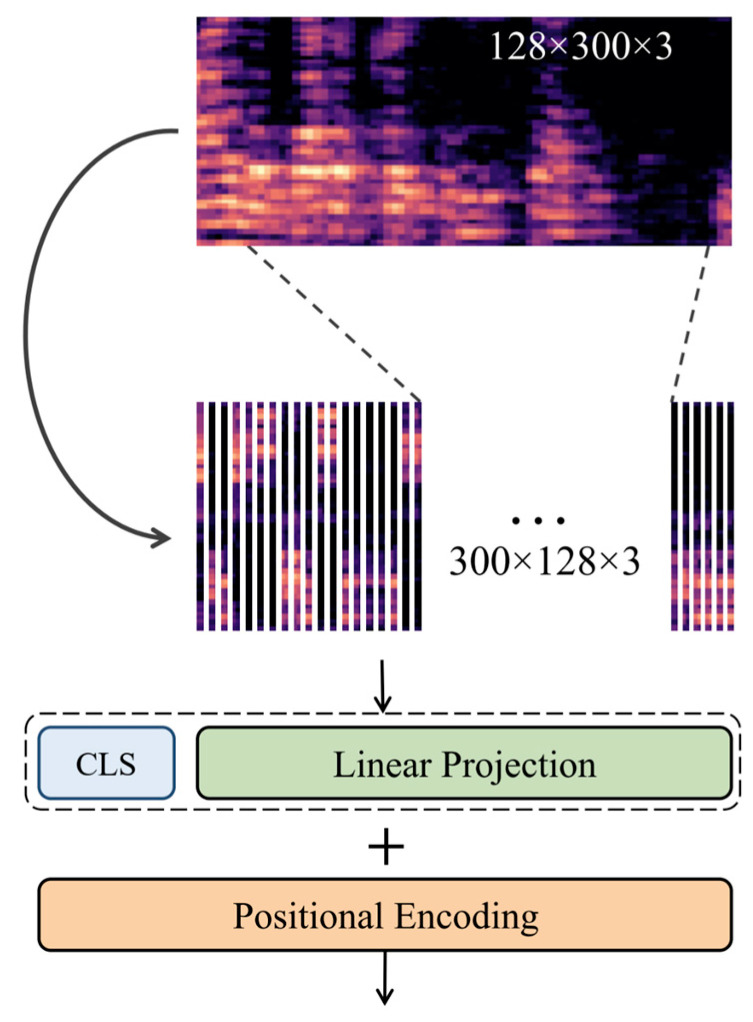
Segmentation of the input MSpec and creation of Temporal Embedding.

**Figure 4 sensors-23-08033-f004:**
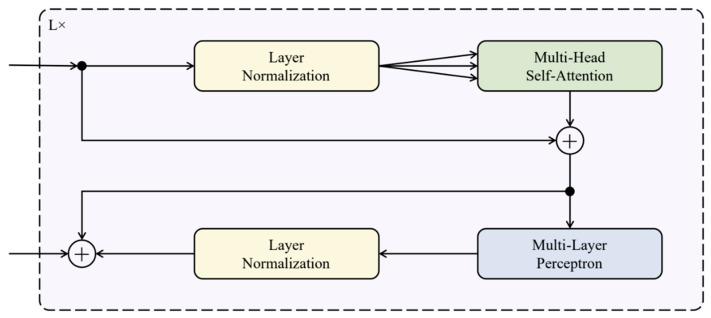
The architecture of Transformer Encoder. L denotes the number of Encoder stacked together.

**Figure 5 sensors-23-08033-f005:**
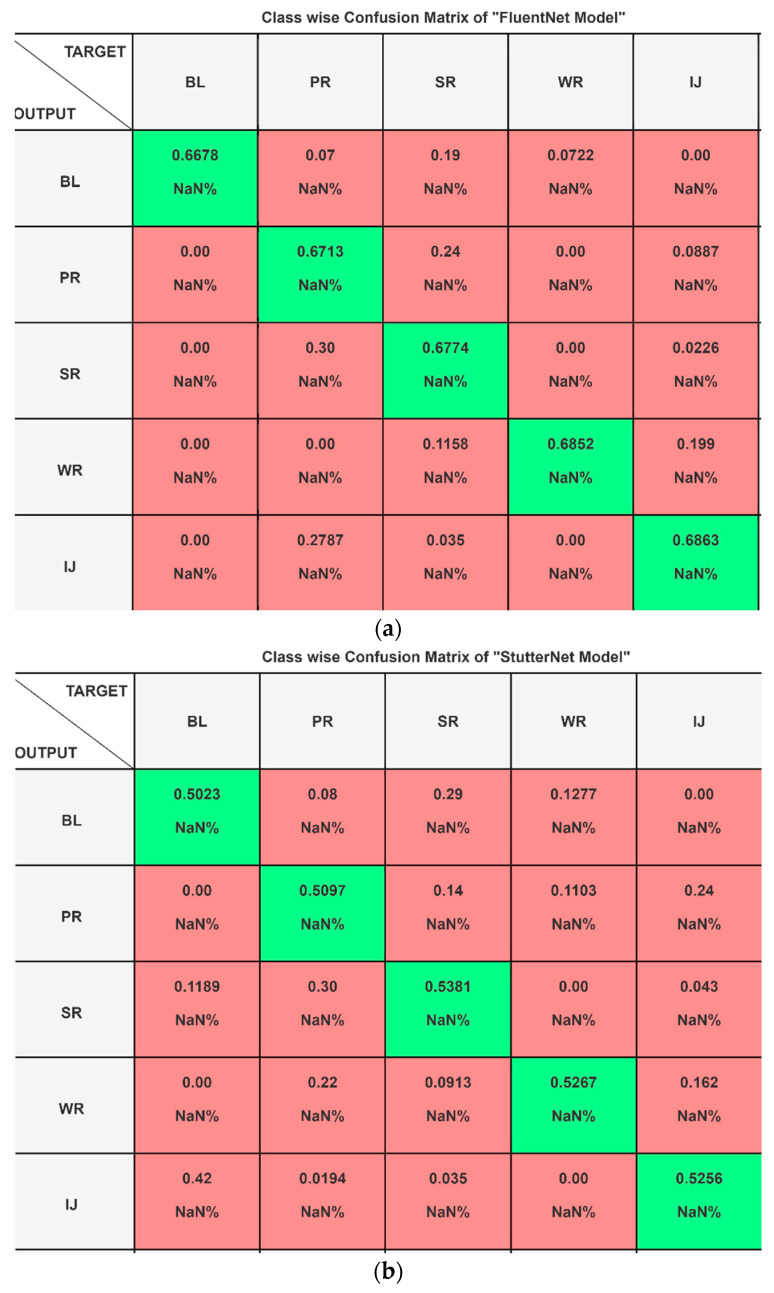
(**a**–**c**) Class wise Confusion Matrix of Three Models Based on the Predicted “OUTPUT” Level vs. Actual “TARGET” Level (“NaN” indicates “Not a Number”). (**a**) Confusion Matrix of “FluentNet Model” Based on the Predicted “OUTPUT” Level vs. Actual “TARGET” Level. (**b**) Confusion Matrix of “ StutterNet Model” Based on the Predicted “OUTPUT” Level vs. Actual “TARGET” Level. (**c**) Confusion Matrix of “TranStutter Model” Based on the Predicted “OUTPUT” Level vs. Actual “TARGET” Level.

**Table 1 sensors-23-08033-t001:** Types of Stuttering Disfluency with Descriptions.

Label	Dysfluency Type	Description
BL	Block	Involuntary pause
PR	Prolongation	Elongated syllable
SR	Sound repetition	Repeated syllable
WR	Word repetition	Repeated word
IJ	Interjection	Added filler word
PhR	Phrase Repetition	Repeated phrase
ND	No dysfluency	Fluent speech

**Table 2 sensors-23-08033-t002:** Summary of Prior Machine Learning Approaches for Stuttered Speech Classification.

Paper	Dataset	Feature	Model/Method	Results
[16]	UCLASS	LPCC	k-NN and LDA	Acc. 89.27% for k-NN and 87.5% for LDA
[18]	Custom	MFCC	SVM	Avg. Acc. 94.35%
[19]	Custom	MFCC	Euclidean Distance	Avg. Acc. 87%
[20]	Custom	MFCC	GMM	Avg. Acc. 96.43%
[21]	KSoF	OpenSMILE and wav2vec 2.0	SVM	Avg. F1 48.17%

**Table 3 sensors-23-08033-t003:** A Summary of Previous Deep Learning Methods for Stuttered Speech Classification.

Paper	Dataset	Feature	Model/Method	Results
[25]	Custom	Respiratory Bio-signals	MLP	Acc. 82.6%
[26]	UCLASS	Spectrogram	ResNet + Bi-LSTM	Avg. Acc. 91.15%
[31]	UCLASS + LibriStutter	Spectrogram	FluentNet	Avg. Acc. 91.75% and 86.7%
[34]	In-house Chinese Corpus	Word and Position Embedding	CT Transformer	F1 70.5%
[37]	UCLASS	MFCC	StutterNet	Acc. 50.79%

**Table 4 sensors-23-08033-t004:** Dysfluent and Non-dysfluent Labels in SEP-28k and FluencyBank.

Dysfluent Labels	Non-Dysfluent Labels
Block	Natural pause
Prolongation	Unintelligible
e Sound Repetition	Unsure
Word Repetition	No Speech
Interjection	Poor audio quality
No disfluencies	Music

**Table 5 sensors-23-08033-t005:** Results of TranStutter trained and tested on SEP-28k and FluencyBank.

	SEP-28K		FluencyBank	
	F1	Acc	F1	Acc
BL	82.63	84.32	77.98	79.46
PR	83.04	85.09	78.14	79.93
SR	86.43	89.61	78.89	82.25
WR	85.19	87.43	80.21	81.82
IJ	85.57	87.91	80.75	81.57
Total	84.06	88.1	78.82	80.6

**Table 6 sensors-23-08033-t006:** Comparative Results in F1 Score and Accuracy on the Combined Dataset.

	FluentNet		StutterNet		TranStutter	
	F1	Acc	F1	Acc	F1	Acc
BL	66.17	66.78	49.26	50.23	84.68	85.1
PR	66.42	67.13	50.2	50.97	85.15	85.76
SR	67.24	67.74	52.49	53.81	87.62	89.57
WR	67.26	68.52	51.34	52.67	86.19	87.38
IJ	67.41	68.63	51.27	52.56	86.43	87.68
Total	66.8	67.76	51.39	52.048	86.82	87.098

## Data Availability

The data that support the findings of this study are openly available at https://github.Com/apple/mL-stuttering-events-dataset/ (accessed on 20 February 2023).

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
