# Peer review of "TranStutter: A Convolution-Free Transformer-Based Deep Learning Method to Classify Stuttered Speech Using 2D Mel-Spectrogram Visualization and Attention-Based Feature Representation"

_sensors, 2023, doi:10.3390/s23198033_

Round 1
Reviewer 1 Report
The paper introduces "TranStutter," a novel deep-learning model based on the Transformer architecture aimed at classifying stuttering in speech. The approach focuses on analyzing disfluent speech patterns characteristic of stuttering. The model's performance is evaluated using the SEP-28k dataset, consisting of audio recordings of People Who Stutter (PWS). The results indicate that TranStutter surpasses existing models in classification accuracy, achieving an impressive 88.1%.
Drawbacks:
1. The paper does not delve into the interpretability of the TranStutter model. Given the clinical implications of the model, understanding how it makes decisions is crucial for its adoption in real-world scenarios.
2. The current version of TranStutter does not seem to be optimized for real-time applications. This limitation could hinder its use in on-the-spot diagnosis or real-time monitoring of stuttering during therapy sessions.
3. While the SEP-28k dataset is substantial, the paper does not specify the diversity of the speech samples. A more diverse dataset would provide a more comprehensive evaluation, including various age groups, ethnicities, and stuttering severities.
4. The paper mentions the potential of extending the work for Multi-Label Classification, given that different stuttering types can coexist in a single audio recording. However, this aspect is not explored in the current study.
Recommendations:
1. Future iterations of TranStutter should focus on improving its interpretability. Techniques like SHAP (SHapley Additive exPlanations) or LIME (Local Interpretable Model-agnostic Explanations) can illuminate the model's decision-making process.
2. To increase the practicality of TranStutter in clinical settings, efforts should be made to develop a real-time version of the model. This would allow instant feedback during therapy sessions and immediate diagnosis.
3. To ensure the model's robustness and generalizability, it should be trained and tested on more diverse speech samples. This includes recordings from various demographic groups and different stuttering intensities.
4. Given the complexity of stuttering, where multiple types can be present simultaneously, it is essential to adapt TranStutter for Multi-Label Classification. This would enhance its accuracy in real-world scenarios with common mixed stuttering types.
5. To ensure the model's clinical relevance, it would be beneficial to collaborate with speech therapists and clinicians. Their insights can guide model improvements and ensure their applicability in therapeutic settings.
6. The abstract of the paper should be expanded.
"TranStutter" presents a promising approach to stuttering classification, leveraging the power of the Transformer architecture. While it showcases notable performance improvements over existing models, some areas, as highlighted above, require attention for its broader adoption. With the recommended enhancements, TranStutter has the potential to revolutionize the diagnosis and treatment of stuttering.
Author Response
Reviewer #1:
- Drawbacks: The paper does not delve into the interpretability of the TranStutter model. Given the clinical implications of the model, understanding how it makes decisions is crucial for its adoption in real-world scenarios. Recommendations: Future iterations of TranStutter should focus on improving its interpretability. Techniques like SHAP (SHapley Additive exPlanations) or LIME (Local Interpretable Model-agnostic Explanations) can illuminate the model's decision-making process.
Response: Thank you for the insightful comments. We have added relevant content in Section 6 diving into the interpretability of the model giving clinical implications and incorporating techniques like SHAP and LIME in the highlighted portions of Section 6.
- Drawbacks: The current version of TranStutter does not seem to be optimized for real-time applications. This limitation could hinder its use in on-the-spot diagnosis or real-time monitoring of stuttering during therapy sessions. Recommendations: To increase the practicality of TranStutter in clinical settings, efforts should be made to develop a real-time version of the model. This would allow instant feedback during therapy sessions and immediate diagnosis.
Response: Thank you for the insightful comments. In Section 5.3, we have added some ways to increase the practicality of real-time applications thereby upgrading the model. Since it can be considered as a limitation, we have addressed the challenges faced to overcome them and how we can increase the practicality of our model in clinical settings.
- Drawbacks: While the SEP-28k dataset is substantial, the paper does not specify the diversity of the speech samples. A more diverse dataset would provide a more comprehensive evaluation, including various age groups, ethnicities, and stuttering severities. Recommendations: To ensure the model's robustness and generalizability, it should be trained and tested on more diverse speech samples. This includes recordings from various demographic groups and different stuttering intensities.
Response: Thank you for the insightful comments. In response to this comment, we would like to bring to your notice that we have already added a section 4.1.2 FluencyBank which is another - more diverse database thereby providing more comprehensive support to our proposed model making it more robust and generalizable.
- Drawbacks: The paper mentions the potential of extending the work for Multi-Label Classification, given that different stuttering types can coexist in a single audio recording. However, this aspect is not explored in the current study. Recommendations: Given the complexity of stuttering, where multiple types can be present simultaneously, it is essential to adapt TranStutter for Multi-Label Classification. This would enhance its accuracy in real-world scenarios with common mixed stuttering types.
Response: Thank you for the insightful comments. We have added relevant content in Section 6 in the highlighted portion about elaborating on and exploring Multi-Label Classification.
- To ensure the model's clinical relevance, it would be beneficial to collaborate with speech therapists and clinicians. Their insights can guide model improvements and ensure their applicability in therapeutic settings.
Response: Thank you for the insightful comments. We have added a section - 5.4 that dives into the model’s clinical relevance. It elaborates on the ways the researchers have adopted to improve the model and ensure its applicability in real-life situations.
- The abstract of the paper should be expanded.
Response: Thank you for the insightful comments. We have updated the abstract and elaborated on it.
Reviewer 2 Report
This research has presented classification model “TranStutter” based on Attention. The proposed methodology is discussed in details as, feature extraction , patch embedding, attention , position encoding, layer normalization. And at last provide evaluation results on datatsets: SEP-28K and FluencyBank . The results show significant improvements.
Suggestions: As the 21% and 36% absolute improvements are huge, hope the author will provide discussions about the reasons. So, could it be the embedding or the attention? which part mostly contributed to the improvement?
And the next one is the 2D spectrum needs a little more clarification.
Author Response
Reviewer #2:
- As the 21% and 36% absolute improvements are huge, I hope the authors will provide discussions about the reasons. So, could it be the embedding or the attention? Which part mostly contributed to the improvement? Section 5.2 Comparative Assessment
Response: Thank you for the insightful comments. We have added the necessary explanations to the ambiguity highlighted in the comments. We have given detailed information about the 2 distinct embedding methods used and also elaborate explanations catering to the huge improvements.
- And the next one is that the 2D spectrum needs a little more clarification.
Response: Thank you for the comments. We have added a few paragraphs highlighted in yellow in Section 3 and 3.1 giving elaborate explanations and clarifications that are required regarding the comment on the 2D Spectrum.
Reviewer 3 Report
I have reviewed your work titled "TranStutter: A Convolution-free Transformer-based Deep Learning Method to Classify Stuttered Speech Using 2D Mel-Spectrogram Visualization and Attention-based Feature Representation" in detail. The major points that I see missing in the study are listed below.
The abstract section should include why the study was done, the innovations of the proposed model and its contribution to the literature, and briefly the results obtained. The abstract part should be rewritten. Information should be given about the classes and data numbers in the data sets. The proposed model should be detailed with a figure and a subtitle should be added. The number of data used in the study and the number of features should be highlighted. "Diagnosis of Heart Diseases Using Heart Sound Signals with the Developed Interpolation, CNN, and Relief Based Model." You can review the related study for the shortcomings I have mentioned. It is not possible to say that there are very important deficiencies until the Result section. The result section consists of approximately one page and 2 tables. The relevant section should definitely be expanded. Confusion matrices, AUC-ROC curves, performance evaluation metrics, etc. should be added. The limitations of the study should be mentioned. As a result, researchers should devote time, especially to the Result section.
Spelling and grammatical errors should be reviewed.
Author Response
Reviewer #3:
- The abstract section should include why the study was done, the innovations of the proposed model and its contribution to the literature, and briefly the results obtained. The abstract part should be rewritten. Information should be given about the classes and data numbers in the data sets.
Response: Thank you for the comments, We have rewritten the abstract which now includes the purpose of the study, the innovations of the proposed model, its contribution to the literature, and the results obtained as well.
- The proposed model should be detailed with a figure and a subtitle should be added. The number of data used in the study and the number of features should be highlighted. "Diagnosis of Heart Diseases Using Heart Sound Signals with the Developed Interpolation, CNN, and Relief Based Model." You can review the related study for the shortcomings I have mentioned. It is not possible to say that there are very important deficiencies until the Result section.
Response: We appreciate your valuable suggestions, and we have made the necessary revisions to address this concern. Our proposed model is stated in Fig-1 with the relevant descriptions for clear understanding. we have already added a section 4.1.2 FluencyBank which is another - more diverse database thereby providing more comprehensive support to our proposed model making it more robust and generalizable.
- The result section consists of approximately one page and 2 tables. The relevant section should definitely be expanded. Confusion matrices, AUC-ROC curves, performance evaluation metrics, etc. should be added.
Response: Thank you for the comments, we have updated the paper and now the Results section i.e. Section 5 is more than 3 pages. We have given a brief overview as well as an elaborate explanation of all our findings throughout our research. We have added a figure (Figure 5) that is a confusion matrix that caters to the comment. A separate table - Table 6 exists in Section 5.2 that contains the performance evaluation metrics like F1 Score and ACCURACY for FluentNet, StutterNet, and TranStutter.
- The limitations of the study should be mentioned. As a result, researchers should devote time, especially to the Result section.
Response: Thank you for the comments, We have updated the paper and added a section that mentions some of the limitations along with how the researchers overcome those limitations by addressing those challenges one by one in Section 5.3.
- Spelling and grammatical errors should be reviewed.
Response: Thank you for the comments, we have updated the entire paper and corrected all grammatical and spelling errors.
Round 2
Reviewer 3 Report
Thank you for your careful revision in the previous round. It is clear that there are serious improvements in the article. There is a serious issue in the article that bothers me. I have serious concerns about the confusion matrix structure given in Figure 5. In confusion matrices, classes should be given for each model and how many data should be predicted correctly or incorrectly in each class should be presented. I don't know how correct it would be to accept the shape you presented as a confusion matrix. Please review this section. Please review the article I mentioned in the previous revision, you can use it as a reference in your work. Additionally, adding a paragraph of information about the other models you used in the study will increase the quality of the study. Thank you.
Best regards.
Reviewing spelling and grammatical errors will increase the quality of the paper.
Author Response
Thank you for your insightful comments and suggestions. In response to your feedback, we've addressed the complexity of Figure 5 by presenting a class-wise confusion matrix of the three models. These matrices now provide a clear comparison between the predicted "output" and the actual "target" levels. We invite you to review figures 5(a), 5(b), and 5(c) for these changes.
Additionally, we've augmented the corresponding descriptions in the manuscript to align with these updates. Your recommended reference proved valuable, and we have duly incorporated it as reference no. [53] in our work.
Your feedback has been instrumental in enhancing the quality of our manuscript. We're deeply appreciative of this.
Furthermore, we have undertaken another round of proofreading to address and rectify any grammar errors present.
Warm regards,
Hsien-Tsung Chang
Round 3
Reviewer 3 Report
Thank you for your successful revision.
.